# A Cross between Bread Wheat and a 2D(2R) Disomic Substitution Triticale Line Leads to the Formation of a Novel Disomic Addition Line and Provides Information of the Role of Rye Secalins on Breadmaking Characteristics

**DOI:** 10.3390/ijms21228450

**Published:** 2020-11-10

**Authors:** Francesco Sestili, Benedetta Margiotta, Patrizia Vaccino, Salvatore Moscaritolo, Debora Giorgi, Sergio Lucretti, Samuela Palombieri, Stefania Masci, Domenico Lafiandra

**Affiliations:** 1Department of Agriculture and Forest Sciences, University of Tuscia, 01100 Viterbo, Italy; francescosestili@unitus.it (F.S.); palombieri@unitus.it (S.P.); masci@unitus.it (S.M.); 2Institute of Biosciences and Bioresources of the National Research Council, 70126 Bari, Italy; benedetta.margiotta@ibbr.cnr.it; 3CREA Research Centre for Cereal and Industrial Crops, 13100 Vercelli, Italy; patrizia.vaccino@crea.gov.it; 4CREA Research Centre for Engineering and Agro-Food Processing, 00189 Rome, Italy; salvatore.moscaritolo@crea.gov.it; 5ENEA, CASACCIA Research Center, Laboratory Biotechnologies, 00189 Rome, Italy; debora.giorgi@enea.it (D.G.); sergio.lucretti@enea.it (S.L.)

**Keywords:** rye, secalins, glutenins, chromosome rearrangements, dough quality

## Abstract

A bread wheat line (N11) and a disomic 2D(2R) substitution triticale line were crossed and backrossed four times. At each step electrophoretic selection for the seeds that possessed, simultaneously, the complete set of high molecular weight glutenin subunits of N11 and the two high molecular weight secalins of rye, present in the 2D(2R) line, was carried out. Molecular cytogenetic analyses of the BC_4_F_8_ generation revealed that the selection carried out produced a disomic addition line (2n = 44). The pair of additional chromosomes consisted of the long arm of chromosome 1R (1RL) from rye fused with the satellite body of the wheat chromosome 6B. Rheological analyses revealed that the dough obtained by the new addition line had higher quality characteristics when compared with the two parents. The role of the two additional high molecular weight secalins, present in the disomic addition line, in influencing improved dough characteristics is discussed.

## 1. Introduction

Wheat represents one of the most important crops for global food security, providing 20% of the total calories and 21% of the daily dietary protein consumed by the human population [1]. The global production has reached about 735 million tonnes in 2018, making the wheat the third most important crop after maize and rice (http://www.fao.org/faostat/en/#data/QC/visualize).

The most cultivated wheat species is *Triticum aestivum*, a hexaploid species with genome AABBDD (2n = 6X = 42) usually known as bread wheat. The success of wheat depends on its adaptability to different climatic conditions (from cold countries as Scandinavia and Russia to Argentina, including elevated regions in the tropics and subtropics) [2] and the unique visco-elastic properties of wheat doughs that permit to produce a vast range of end products, unmatched by the other two major cereals maize and rice, such as different kinds of bread, pasta, noodles, pizza, cakes, snacks, and other bakery foods [3].

Current global genetic gains in wheat have been reported to be around 1% per annum which is insufficient to meet the future demand in the face of a constantly increasing world population [4]. Modern wheat varieties are affected by a reduced genetic diversity, as consequence of different processes such as polyploidization, domestication, and modern plant breeding, the need to introduce new genetic diversity into breeding programs, to satisfy the demand of varieties with good yield and superior processing and nutritional characteristics, represents a primary objective, also in light of the new challenges posed by the global climate change [5].

Wheat, along with rye and barley, is part of the *Triticeae* tribe which includes approximately 325 annual and perennial species [6], forming a large germplasm reservoir of genes that can be used for cereal crop improvement [7,8].

Wide hybridization represent an important approach to introduce desirable alien genes from distant relatives in wheat and the first step in this endeavor is represented by the production of wheat–alien chromosome addition or substitution lines These lines are used as a bridge to generate useful translocations lines and many and numerous such lines have been produced using barley, rye, and several species of the genus *Dasypyrum*, *Aegilops, Agropyron*, and *Thinopyrum* [9,10,11,12].

In order to produce useful translocations several methodologies are available and have been described in many papers, such as use of *Ph* gene mutants to induce homoeologous recombination, use of radiation or gametocidal gene, transgenic approach [11,13,14,15].

Rye represents an important source of useful alien genes for wheat improvement, and at this regard the intergeneric introgressions of the Robertsonian 1BL/1RS wheat-rye translocation represents one of the most successful achievements, as also witnessed by the high number of wheat variety released worldwide [16]. In this material the short arm of wheat chromosome 1B is replaced by its counterpart from the rye genome which arose spontaneously from centromeric breakage reunion and fusion of two chromosome arms broken at the centromere conferring to wheat resistance to several diseases, high yield performance, wide adaptation, resistance to drought, increasing amount of micronutrient, protein content, and arabinoxylan content [17,18,19].

Results of several studies carried out on wheat-rye substitution, addition lines, triticale, and their progeny from crosses between triticale and wheat have demonstrated that deletions and translocations of chromosomal regions and chromosome arms are among the most common spontaneous chromosomal changes generated. In addition to these rearrangements, formation of minichromosomes and chromosomes with multiple centromeres have also been observed, as well as elimination of most or all of the D genome chromosomes [20].

In some cases, the composition of the gluten proteins, and particularly of the high molecular weight glutenin subunits (HMW-GS), was modified as consequences of gene deletions, silencing, or coding gene sequence rearrangements [21,22,23], resulting in the generation of new genetic diversity in this group of proteins.

Wang et al. [24], determined the HMW-GS composition of a large number of Chinese bread wheat varieties, using SDS-PAGE. Within the material analyzed, a novel pattern of HMW-GS was detected in Xiaoyanmai 7 (XY7), a variety grown in the Southern regions of China. Six possible subunits seemed to be present, one of which showed the same mobility as subunit 21, normally present in some bread wheat varieties carrying the 1B.1R translocation. Although Wang et al. [24] suggested that some of the unusual subunits could have been introduced from *Agropyron elongatum,* used during the breeding program in which XY7 was obtained, no clear identification about the source of the novel subunits was achieved.

Subsequently, in order to characterize the HMW-GS present in XY7, Feng et al. [25] carried out an in depth study using biochemical, molecular, and cytogenetical approaches.

In particular, SDS-PAGE separation indicated that XY7 possessed four HMW-GS, two of which resembled the pair of HMW-GS 7 + 8, encoded by genes present at the *Glu-B1* locus, whereas the other two subunits could not be classified. Reversed phase high performance liquid chromatography (RP-HPLC) along with N-terminal sequence and Mass Spectrometry (MS) analyses of the two unknown subunits confirmed that they were indeed HMW-GS. Degenerated primers were used to amplify all the HMW-GS genes of XY7 and deduced amino acid sequences showed that the two novel subunits corresponded to x- and y-type HMW secalins, with the first smaller than the latter and normally associated with genes present on the long arm of the 1R chromosome at the *Sec-3* locus.

Further cytogenetical analysis demonstrated that XY7 was a wheat-rye secondary substitution line in which six pairs of rye chromosomes had replaced their homoeologs of the D genome, except the pair of the 2D chromosomes.

Simultaneously with the biochemical and molecular characterization of the HMW-GS present in XY7 above described [25], a backcrossing approach was initiated to incorporate the two novel subunits in a line of common wheat (N11), in order to evaluate the effect of the two novel subunits on the rheological characteristics of the doughs. We must point out that the backcrosses began before we realized that XY7 was not a wheat, as resulted from its chromosomal composition.

The work described in this paper reports novel findings of the appearance of large chromosomal rearrangements, resulting from the strong selection pressure exerted during the backcrossing process. The selection based on the presence of the two novel subunits, at each backcross step, resulted in the production of a disomic addition line in which the D genome chromosomes of wheat, had replaced all the R chromosomes of rye; the two extra chromosomes observed consisted of the long arm of chromosome 1R from rye fused with the satellite body of the wheat chromosome 6B.

## 2. Results

### 2.1. Electrophoretic Analysis of Gliadins and HMW-GS

The electrophoretic separation of gliadins and glutenin subunits present in the two parents XY7, N11 and in the derived N11^XY7^ line are reported in Figure 1a,b.

In a previous work we have shown that XY7 can be considered a substituted triticale possessing the set of chromosomes of the A and B genomes, six pairs of rye chromosomes, and the 2D(2R) substitution [25]. The analysis of the gliadin of XY7 shows the absence of the ω components, with slow electrophoretic mobility, encoded by genes present at the *Gli-D1* locus on the short arm of the 1D chromosome, and visible in N11. In addition, the ω-secalins encoded by genes present on the short arm of the 1R chromosomes, at the *Sec-1* locus are present in XY7. The line N11^XY7^ shows an identical gliadin pattern to the line N11 as result of the repeated backcrosses and selection. Analysis of the electrophoretic separation of glutenin subunits reveal major differences in the LMW-GS between XY7 and N11. The absence of the 1D chromosome, in addition to the gliadin, is also responsible for the loss of the LMW-GS associated at the *Glu-D3* locus. On the contrary the LMW-GS of N11^XY7^ result identical to N11. SDS-PAGE separation of the glutenin subunits shows that N11 has HMW-GS 1, 7 + 8 and 2 + 12 encoded by genes present at the *Glu-A1*, *Glu-B1,* and *Glu-D1*, respectively, present on the long arm of the homoeologous group 1 chromosomes. The line XY7 has only the pair of subunits 7 + 8 and two HMW secalin (Ry and Rx) subunits, previously identified [25], encoded by genes present at the *Sec-3* locus. In the XY7 y-type subunit is larger than the x-type subunit, as deduced by DNA sequencing of their corresponding genes, contrary to what is normally observed in bread wheat. The line N11^XY7^ shows seven glutenin subunits, the five HMW-GS of N11 and the two HMW-secalins from XY7.

### 2.2. Cytogenetic Analysis by Genomic In Situ Hybridization (GISH) and Fluorescence In Situ Hybridization (FISH)

The cytogenetic analysis of the N11^XY7^ line revealed a chromosome number of 2n = 44 (Appendix A) instead of the standard bread wheat complement of 2n = 42 chromosomes. In order to better discriminate among wheat and rye chromosomes and identify the additional chromosomes pair, GISH and single and double target FISH experiments were performed on metaphase spreads of N11^XY7^ and parental lines. Genomic hybridization analysis of the N11^XY7^ line (Figure 2a) with total rye DNA, showed a clear hybridization signal on the whole long arm of a chromosomes pair, the candidate additional one. Moreover, a peculiar DAPI band was clearly visible at the telomeres of the same chromosomes pair, which facilitates their identification as a secale-derived translocation in the wheat background (Figure 2a). In XY7 six labeled chromosomes pairs where detected, as expected (Figure 2b).

FISH experiments were performed sequentially after GISH using the (GAA)_7_ oligonucleotide, and the rDNA probe pTa71; both probes have been widely exploited to characterize and discriminate wheat and rye chromosomes [26,27]. Our results showed the parental lines exhibiting a (GAA)_7_ distribution in agreement with previous reports [28], while the N11^XY7^ line disclosed the lack of the satellite body on chromosome 6B and the following lack of the (GAA)_7_ signal localized on the satellite (Figure 3).

To further investigate on satellite chromosome morphology and structure the pTa71 ribosomal probe was used on the two parents XY7, N11, and the derived N11^XY7^ line.

In bread wheat, hybridization signals with the pTa71 probe are usually localized at the secondary constriction of satellites chromosomes pairs 1B and 6B and on chromosomes pair 5D, while in rye pTa71 labeling is present at the secondary constriction of the chromosome 1R [29]. In our analysis the pTa71 probe showed a standard distribution on the parental lines N11 and XY7 (Figure 3); on the N11^XY7^ line the signal was visible on chromosome 5D and at the secondary constriction of chromosome 1B, as expected, but on chromosome 6B, which was lacking both the secondary constriction and the satellite body, the pTa71 was localized at the telomeric region of the short arm. Besides, two centromeric labeling spots were visible on the extra “chromosomes pair” (Figure 3). By comparing GISH and FISH labeling pattern of the parental lines with those shown in N11^XY7^, we hypothesized the translocation of the satellite body from wheat chromosome 6B to the additional rye chromosomes pair present in this new N11^XY7^ line. Our hypothesis explains the presence of the pTa71 hybridization signal at the centromeric region of the additional chromosomes pair and the uncommon localization of pTa71 probe at the telomeric region of the short arm of chromosome 6B which probably, in spite the lack of the large part of the satellite body, still retains some copies of the rDNA genes. In addition, two (GAA)_7_ hybridization spots expected on the 6B satellite body, were present on the “short arm” of the extra chromosomes pair (data not shown).

### 2.3. Qualitative Characterization

The values and means of the qualitative traits for the genotypes under study and growing season are reported in Table 1 and Appendix A. As regards the test weight (TW), the parental lines had contrasting values, very high for line N11 (82.6 kg/hL), and very low for XY7 (69.2 kg/hL). N11^XY7^ showed very good values, in both years, with an average of 80.1 kg/hL, comparable to N11. No significant variation was found among the three lines for the hardness, all them showing hard texture (hardness values ranging from 89 to 100). A significant variation was observed between the growing seasons, with Y1 resulting in an increase of the hardness index by 28% in comparison to Y2. The protein content (PC) of both parental lines was significantly lower than the one observed in N11^XY7^ (14.2% d.m. on average), with no significant differences between the growing seasons. As regards the Gluten Index, N11^XY7^ had a very high value (99.5), significantly higher than N11 (86.5) and XY7 (64). No significant variation between growing seasons was observed. The rheological characteristics, as determined by the alveographic test, showed large differences among the three lines (Figure 4). In this test, line N11^XY7^ showed on average very high extensibility (L: 175 mm), significantly higher than that of N11 (80 mm) and XY7 (55 mm). Dough strength (W) of N11^XY7^ was twice that of the best parent, N11, and four times higher than that of XY7. The differences were confirmed in both growing seasons, the differences not being significant between Y1 and Y2 for all the alveographic parameters.

As for the Brabender farinographic test, the analyses showed that, in spite of its high protein content, N11^XY7^ water absorbance was similar to that of N11 and slightly higher than that of XY7 (60.9% vs. 60.8%, and 54.1%, respectively). In comparison to the parental lines, N11 and XY7, N11^XY7^ had significantly higher development time (9.7 min vs. 2.8 min and 1.4 min, respectively) and dough stability (11.8 min vs. 5.3 min and 3.6 min, respectively). The degree of softening was similar for N11 and N11^XY7^ (62 and 51 BU, respectively), while XY7 had a significant higher value (119 BU). The FQN, as expected, was highly correlated with the dough development time, and was significantly higher in N11^XY7^ (171 mm) than in N11 (62 mm) and XY7 (50 mm). For all the farinograph parameters, no significant differences were observed between growing seasons.

## 3. Discussion

### 3.1. Chromosomal Rearrangements and Their Origins

Genomic in situ hybridization (GISH) is a widely used approach for detecting alien chromatin in wheat genetic background [30]. The alien DNA can be further analyzed by fluorescence in situ hybridization (FISH) using repeated sequences as probes [31]. Among the different classes of repeated sequences, ribosomal DNA (rDNA) and simple sequence repeats (SSR) are the most valuable cytological markers in discrimination of plant chromosomes, and they have been widely used in rye and wheat cytogenetic analysis. In this work a new bread wheat line (N11^XY7^) derived from the cross between the bread wheat line N11 and XY7, with the latter harboring six pairs of chromosomes of rye [25], have been characterized by means of GISH analysis on N11^XY7^ metaphase using total rye DNA as probe, thus enabling the identification of an additional chromosome pair to the bread wheat standard complement.

Further biochemical analysis and the cytogenetic chromosomes characterization by FISH labeling using rDNA and the SSR (GAA)_7_ revealed that the additional pairs was, in fact, composed of the whole long arm of chromosome 1 of rye (1RL) fused with the satellite body derived from the chromosome 6B of wheat (6B SAT).

The presence of chromosomal rearrangements in the crosses between wheat × wild relatives has been particularly observed and described in several studies conducted on wheat-rye substitution and addition lines, triticale and their progeny from crosses triticale × wheat [32,33,34].

McClintock [35] pointed out that major restructuring of chromosome components may arise in wide species crosses and continue to arise in its progeny, as consequences of ‘genome shock’.

These processes are associated with translocations, inversions, deficiencies, duplications with new stable or relatively stable “species” or “genera” derived from such initial hybrids. Triticale is described as an example by McClintock [35] in which the combined set of chromosomes of wheat and rye are not entirely stable and further selection in subsequent generations does not completely eliminate this instability which resulted from the loss of parental bands in triticale or, conversely, novel bands appearing in the allopolyploid that were not seen in the parental genomes.

Tang et al. [34] reported that wheat chromosomal alterations could be easily induced by rye chromosomes in wheat-rye hybrids, suggesting that the stability of the wheat genome could be effectively suppressed by the R genome.

Lukaszewky and Gustafson [36] carried out a large number of cytological analyses of four triticale X wheat populations and found that among 785 karyotyped plants 195 wheat/rye and 64 rye/rye translocated chromosomes were present, in addition modification of rye chromosomes by deletion or amplification of telomeric heterochromatin was identified. Out of 39 identified wheat/rye translocations, 10 occurred between homoeologous and 29 between nonhomoeologous chromosomes, five involved A-genome chromosomes, six B-genome chromosomes, and the remaining 28 involved D-genome chromosomes. It has been suggested that very likely most of the translocations described resulted from mis-division of univalents at meiosis and subsequent random fusion of telocentric chromosomes [36].

Darlington [37] suggested that univalents in meiosis have a tendency to mis-divide at the centromeres, producing telocentric chromosomes. The 1RS/1BL translocations, a well-known example of the use of alien chromosomes in wheat breeding, are centric translocations formed by mis-division followed by the fusion of the mis-division products, in which the short arm of wheat chromosome 1B is replaced by the short arm of 1R rye chromosome [38]. Fu et al. [32,33] observed that even single alien rye chromosome added to wheat genome could induce alterations of wheat and rye chromosomes. They investigated the alterations and abnormal mitotic behaviors of wheat chromosomes, induced by wheat-rye 4R, 6R, and 7R monosomic addition lines [32] and by wheat-rye 1R monosomic addition lines [33]. In the addition lines containing the whole 1R chromosome or 1R chromosome arms and in their self-progeny lagging chromosomes, micronuclei, and chromosomal bridges were observed.

Beside centromere (primary constriction), the secondary constrictions may represent a possible region of breakage. It is known, in fact, that chromosome constrictions are points of weakness in the chromosome [39]. In addition, secondary constriction often coincides with the nucleolar organized region (NOR) containing the rDNA gene. The 45S rDNA in plant is considered a fragile site [40], that is a large and highly sensitive region which is inclined to form abnormal poor-staining lesions with gaps, constrictions, or breaks in metaphase chromosomes.

In our lines, the mitotic instability induced by the presence of rye chromosome in wheat background join to the chromosome secondary constriction “weakness” might have led to the centromeric breakage of chromosome 1R followed by the fusion of its long arm with the satellite body of the wheat chromosome 6B. A similar translocation line was observed by Badaeva et al. [41] in two related strains of French tetraploid triticale. The translocation line indicated as T6A:1B-1, was formed by the break that occurred in the short arm of chromosome 6A near the centromere and the second break within the region of secondary constriction of chromosome 1B, which resulted in the emergence of a new satellite chromosome formed by the long arm of 6A and the satellite of 1B [41].

Already in 1962, Fergusson-Smith [42] observed that in human metaphase chromosomes with secondary constrictions are close to one another more often than would be expected from mere chance distribution. This phenomenon has been termed satellite association. Rye chromosome 1R and wheat chromosome 6B are both satellite chromosome holding secondary constriction, their possible proximity in the nucleus could explain why the fusion occurred among these two nonhomoeologous chromosomes. On the other hand, nonhomeologous wheat/rye translocation have been previously reported [36].

Changes in gluten protein composition, as consequences of gene deletions, silencing, or coding gene sequence rearrangements have also been reported by different authors. Two hexaploid lines of triticale originated from the cross between the bread wheat cultivar M8003 and Austrian rye (*Secale cereale* L.) were used by Li et al. [23] to evaluate morphologic traits, chromosome composition, and storage protein composition. The two triticale lines differed for plant height, spike length, and number of spikelets. Cytological analyses showed the complete elimination of the D-genome chromosomes and the presence of major chromosomal rearrangements, involving chromosomes of the A and B genomes in both triticales. In addition, large variation in HMW-GS, LMW-GS, and secalins between the two triticales was detected. At last, novel patterns that distinctly differentiated the two triticale lines from the parents also appeared, indicating that coding sequence rearrangements, activation of novel gene expression, gene silencing, or deletions could be responsible for the changes observed.

Yuan et al. [21] analyzed by SDS-PAGE the HMW-GS composition of F_1_ and F_2_ progeny of a cross between a Japanese bread wheat landraces and rye. Results of their analyses revealed absence and/or additional novel HMW-GSs in the endosperm of F_1_ and F_2_ hybrid seeds. Analysis of cloned sequences of HMW-GS revealed that deletions involving the in-frame stop codon, present in the *Glu-Ax1* null allele of the bread wheat parent, had occurred. This produced a novel active *Glu-Ax1* allele in some F_1_ and F_2_ plants corresponding to one of the novel subunits detected by SDS-PAGE, as demonstrated by heterologous expression in *E. coli*. The results of this work lead Yuan et al. [21] to suggest that mitotic illegitimate recombination between two copies of a short repeat sequence had resulted in the observed deletions with consequent change in HMW-GS compositions.

In a different work Yuan et al. [22], in order to investigate the possibility of using a wheat-rye hybridization method for inducing de novo phenotypes, crossed the bread wheat cultivar Shinchunaga, possessing the very large HMW-GS 1Dx2.2 associated with the *Glu-D1* locus, with the Chinese rye landrace Qinling. Electrophoretic analysis of HMW-GS of F_2_ seeds, together with PCR amplification of the HMW-GS genes, their cloning and sequencing permitted to identify a subunit, designated Glu-1Dx2.2^v^ derived from the *1Dx2.2* gene through deletion of a direct repeat of 295 bp length and an intervening sequence of 101 bp present in its central repetitive region.

Additionally, in this case, Yuan et al. [22] postulated that illegitimate recombination was, very likely, the mechanism responsible for the appearance of the new subunit Glu-1Dx2.2^v^.

Differently from the changes above reported, our work proves that the pressure exerted during the backcrossing process resulted in the production of a disomic addition line in which the pair of the novel chromosomes are the result of major rearrangements.

### 3.2. Characteristics of HMW Secalins and Their Possible Role on Rheological Properties of Doughs

Glutenin polymers are the major factor responsible of dough elasticity and strength and are formed by HMW-GS and low molecular weight glutenin subunits (LMW-GS), held together by intermolecular disulfide bonds. The existence of highly positive correlation between large sized polymers as measured by the relative amount of unextractable proteins (%UPP) and dough processing properties has been investigated by many authors and it has been demonstrated that the HMW-GS play a major role compared to LMW-GS [43,44,45,46].

As far as HMW-GS is concerned, intensive genetic research has revealed that, in hexaploid wheat, HMW-GS are encoded by orthologous gene sets, present at the *Glu-A1*, *Glu-B1,* and *Glu-D1* loci located on the long arms of the homoeologous group 1 chromosomes [47]. Each locus contains two tightly linked paralogous genes encoding two types of HMW glutenin subunits, designated as x- and y-type subunits, with higher and lower molecular weight, respectively [48].

Usually, the number of cysteine residues in x and y type subunits are four (three in the N-terminal domain and one in the C-terminal domain) and seven (five in the N-terminal domain, one in the repetitive domain, and one in the C-terminal domain), respectively [48].

These cysteine residues form disulfide bonds within and between subunits, which are important in determining structure and properties of the gluten complex in wheat dough [48].

Allelic variation has been identified at each *Glu-1* locus with both number and type of subunits impacting breadmaking quality. In particular dough mixing and baking characteristics improved when the number of subunits increased from three to five [47,49], thanks to their contribution to the formation of large-sized glutenin polymers, as deduced by the proportion of %UPP [44,50].

Gupta and MacRitchie [51] reported that the genotypes with the pair of subunits 5 + 10, associated at the *Glu-D1* locus, had a significant higher size distribution of polymeric glutenins compared to those with the pair of allelic subunit 2 + 12, suggesting that the differences in number of cysteine residues between 1Dx5 (5 cysteines) and 1Dx2 (4 cysteines) was responsible for the superior characteristics associated to varieties possessing the pair of subunits 5 + 10.

Pirozi et al. [52], compared quality characteristics of two biotypes of the bread wheat variety Avocet differing only in the HMW-GS composition. Avocet A possessed HMW-GS composition: null (*Glu-A1*), 7 + 8 (*Glu-B1*), 5 + 10 (*Glu-D1*), whereas HMW-GS composition of Avocet C was 1, 20x + 20y, 5 + 10; Avocet A exhibited an increase in polymeric protein, a decrease in the gliadin-to-glutenin ratio, and a marked decrease in %UPP. Avocet C doughs exhibited greater extensibility and shorter Mixograph, dough development times, and baked into smaller loaves compared to those from Avocet A, in spite of the fact that, compared to the former, the latter possesses an additional HMW-GS (subunit 1 at the *Glu-A1* locus).

The presence of the additional chromosomes pair in N11^XY7^ and associated HMW secalins resulted in an unexpected rheological behavior, but also in an exceptionally high technological quality of the dough, which allows N11^XY7^ to be classified as improver wheat in Italy, excellent or class E wheat in France and Germany, hard red wheat in the USA. Moreover, in spite of its very high dough strength, N11^XY7^ keeps a very good extensibility, which is one of the main objectives of bread wheat quality improvement. Usually, in fact, strong gluten often suffers for low extensibility, which is a serious problem, since adequate extensibility is required for proper dough handling and baking performance [53].

Extensibility is highly influenced by gliadins and protein content [54,55,56], but the gliadin pattern of N11^XY7^ is not different from the pattern of the parent N11, thus excluding gliadins to be responsible for this change. In addition, in our opinion, the extensibility of N11^XY7^ seems too high to be justified only by the result of the higher protein content found in the line.

The farinographic behavior of N11^XY7^ was quite peculiar, as well. In spite of its higher protein content, in comparison to both parents, N11^XY7^ did not show a proportional increase in water absorbance. On the contrary, great differences were observed in development time, with almost 10 min, on average, necessary to reach optimum dough consistency.

Unlike HMW-GS the role of HMW secalins in influencing breadmaking quality has been addressed in a very limited way. Kipp et al. [57] found that rheological properties of wheat gluten were negatively affected, when purified HMW secalins were incorporated in wheat dough, with a strong reduction in maximum resistance and a slight increase in extensibility. The authors suggested that the observed effects were associated with the more frequent occurrence of cysteine residues in the HMW secalins affecting the formation of large polymeric glutenins.

Amiour et al. [58] analyzed substitution lines 1R(1A), 1R(1B), and 1R(1D), in which the entire 1R chromosome replaces the homoeologous chromosome of the group 1 of wheat, in order to compare the contribution of the HMW secalin subunits, encoded by genes present at the *Sec-3* locus, to that of HMW-GS on the technological characteristics of the doughs.

Rheological studies demonstrated that the contribution of the three substitutions on breadmaking quality was different with the substitution 1R(1A) showing a significant positive effect compared to 1R(1B) and 1R(1D) substitutions. The alleles encoding HMW secalin subunits at the *Glu-R1* had an intermediate value between the null allele and subunit 2^*^ encoded at the *Glu-A1* locus. Negative effects were associated with the 1R(1B) and 1R(1D). It should be emphasized that these results are influenced by the presence of the monomeric ω-secalins and the 40 k group of γ-secalins encoded by genes present at the *Sec-1* locus, on the short arm of the 1R chromosome [59].

A negative effect of the HMW secalins, on wheat quality characteristics, was also reported by Graybosch [60]. This result was obtained comparing lB(1R) substitution lines with the 1BL/1RS sister lines. The presence of the 1RL resulted in further reductions in dough strength parameters compared to 1RS alone.

Agronomic and quality data were obtained by Kumlay et al. [61] using the three translocation lines 1AS/1RL, 1BS/1RL, and 1DS/1RL, in which the long arm of chromosome 1R replaced the long arms of the homoeologous group 1 chromosomes of the bread wheat variety Pavon. This resulted in the introduction of the *Sec-3* locus, carrying genes encoding HMW secalins and elimination of the three different *Glu-1* loci carrying genes for the HMW-GS. Rheological data indicated that among the three translocation lines the 1AS/1RL was better than the control bread wheat variety Pavon carrying the subunit 1Ax1 associated at the *Glu-A1* locus.

Additionally, in our case, the N11^XY7^differs from N11 only for the two additional HMW secalins and the drastic changes in rheological characteristics observed, in addition to the high protein content, must in some way be associated to their presence.

In addition, the sequence of 1Ry gene present in XY7 included eight cysteine residues [25] rather than the seven normally present in y-type subunits in bread wheat, with the additional cysteine located within the C-terminal domain upstream of the one normally present in bread wheat; this cysteine residue has been suggested to be restricted to y-type HMW secalins by De Bustos and Jouve [62].

In bread wheat it has been widely demonstrated that number of subunits in addition to the number and distribution of cysteine residues, available for intermolecular bonds, are responsible for differences of breadmaking quality of different wheat varieties. Results have been obtained that show that additional cysteines do not always influence positively the bread-making quality of flour, as in the case of the 1Dx5 subunit [63,64,65]. On the basis of the rheological data obtained we can hypothesize that the additional cysteine present in the 1Ry subunit could contribute to modulate differently the size distribution of glutenin polymers, producing a dough with a very large extensibility.

## 4. Materials and Methods

### 4.1. Plant Materials

A wheat line was produced by crossing the bread wheat line N11 and the 2D(2R) disomic triticale XY7, followed by four backcrosses with N11 and designated N11^XY7^. The HMW-GS present in the seeds obtained after each backcross were analyzed by one dimensional polyacrylamide gel electrophoresis in SDS (SDS-PAGE), according to Feng et al. [25], and those containing seven HMW-GS subunits (five typical of N11 plus the two HMW secalin subunits from rye) were selected for the next step of backcrossing. The gliadins were analyzed by acid polyacrylamide gel electrophoresis (A-PAGE) as described by Camerlengo et al. [66].

### 4.2. Field Trials

Bread wheat N11, XY7, and BC_4_F_8_ seeds of N11^XY7^ were grown in open field during 2016/2017 (Y1) and 2017/2018 (Y2) growing seasons at the Experimental Farm of the University of Tuscia, located in Viterbo, Italy (lat. 42°26′ N, long. 12°4′ E, altitude 310 m a.s.l.), in 10 m^2^ plot trials, with three replications. The experimental site is characterized by a Mediterranean climate with mean annual maximum and minimum temperatures of 19 and 8 °C, respectively, and annual rainfall of 743 mm. The agronomic practices were those recommended for high yields, including sowing 450 germinating seeds m^−1^, seed dressing, chemical control of weeds, 150 kg/ha of nitrogen applied in three top dressing, no application of growth regulators, fungicides, and pesticides.

After harvesting, test weight (TW) was determined by means of a Dickey-John GAC2000 grain analysis meter (Dickey-John Corp. Auburn, IL, USA), according to the supplied programme.

### 4.3. Qualitative Analyses

Grain samples (50 g) from each plot were ground to wholemeal using a 1-mm-sieve Cyclotec mill (Foss Tecator AB, Höganäs, Sweden). Grain hardness (Ha) (AACC 39–70) [67] was determined on the resulting wholemeal by a NIR System Model 6500 (FOSS NIR Systems, Laurel, MD, USA).

The three replications for each line were bulked and milled with a Bona 4RB (Bona, Italy) experimental mill after tempering according to their hardness. Flour protein content (PC) (N × 5.7, dry matter basis, AACC 39–11) was determined by NIRS. The gluten index was evaluated on flour according to ICC-158 [68] using the Glutomatic System (Perten Instruments, Hagersten, Sweden). All the analyses were repeated twice (technical replicates) for each sample.

The rheological properties of flour were evaluated using a Chopin alveograph, according to ICC-121 and a Brabender farinograph, according to ICC-115-D [68]. Qualitative data were analyzed for significance by one-way ANOVA in combination with Tukey’s HSD (honest significant difference) test for post-hoc comparisons of means, using the JMP software package (SAS Institute Inc., Cary, NC, USA, 1989–2019).

### 4.4. Cell Cycle Synchronization

Wheat (*Triticum aestivum* L.): bread wheat line N11, XY7, and the derived line N11^XY7^ and rye seeds (*Secale cereale* L.) were soaked in aerated water for 8 h and germinated on moistened filter paper for 2–3 days in the dark at 19 °C. Root tips from 1–2 cm long roots were used for further analysis.

In root tips, cell cycle synchronization and accumulation of metaphases was accomplished according to Doležel et al. [69] with minor modifications. Young seedlings were transferred in an aerated Hoagland’s solution [69] at 25 °C and exposed to 1.25 mM hydroxyurea for 18 h, followed by a recovery period of 4 h. After blocking at metaphase by a 2 h treatment with 2.5 µM amiprophos-methyl, roots were incubated overnight in ice water and fixed in 3:1 ethanol-glacial acid. Metaphase spreads for genomic in situ hybridization (GISH) and fluorescence in situ hybridization (FISH) analysis were prepared from synchronized root tips as described by Schwarzacher and Heslop-Harrison [70]. Pretreatments and stringency washes were applied only to the slides hybridized with total rye DNA (GISH) and pTa71 probe (standard FISH) while these steps were omitted in nondenaturing FISH (ND-FISH) with oligonucleotides as probe.

### 4.5. DNA Probes

Both the rye total genomic DNA, used as probe in GISH analysis, and the 18S-5.8S-26S rDNA clone pTa71 [71], probe in FISH experiments, were labeled with Cy3 (Cyanine 3) by nick-translation according to manufacturer’s instructions (Nick Translation Mix, Roche, Basel, Switzerland). The oligonucleotide 5′-FITC-(GAA)_7_-3′-FITC was synthesized and labeled by Eurofins MWG Operon (Ebersberg, Germany).

### 4.6. Nondenaturing FISH ND-FISH

Fast FISH labeling in nondenaturing conditions was performed, using the synthetic oligonucleotides (GAA)_7_ as probes, according to Cuadrado and Jouve [29] with the modifications as described by Giorgi et al. [72]. The hybridization mixture was composed of 1.6 ng/µL of probe DNA in a total volume of 60 µL of 2xSSC (300 mM sodium chloride, 0.3 mM trisodium citrate) per microscope slide. After incubation at room temperature for 1 h the slides were washed for 10 min in 4xSSC with 0.2% Tween20 and the chromosomal DNA were counterstained with DAPI (49.6-diamidino-2-phenylindole) and mounted in an antifade solution (Vectashield, Vector Labs, Burlingame, CA, USA).

### 4.7. FISH

FISH with the probe pTa71 was performed on metaphase spreads according to Schwarzacher and Heslop-Harrison [70]. The hybridization mixture (50 µL) was made of 50% formamide, 2xSSC (0.30 M NaCl, 0.030 M sodium citrate), 10% dextran sulfate, 30 ng mL^−1^ denatured salmon sperm DNA, 100 ng probe and water to the final volume. The mix was spread on slides and DNA was denatured at 80 °C for 5 min in a thermalcycler. Hybridization was carried on overnight at 37 °C in a moist chamber and the most stringent washes were performed twice in 20% formamide in 0.1xSSC at 42 °C for 5 min. After DNA counterstaining with DAPI (0.2 µg mL^−1^), the slides were mounted in Vectashield antifade solution.

### 4.8. GISH

GISH was performed according to Ceoloni et al. [30] with some modifications: wheat blocking DNA was used in a ratio of 20:1 to the probe, consisting of rye genomic DNA labeled by nick translation. Hybridization conditions and the following steps were the same as for FISH with the pTa71 probe.

## 5. Conclusions

Chromosomal rearrangements are frequently observed in *Triticeae* and occur widely in natural and resynthesized polyploids.

The use of molecular markers, sophisticated cytogenetic approaches, and biochemical analysis have greatly contributed to unveil molecular changes following wide crosses and demonstrated that the chromosomal rearrangements following the polyploidization processes, such as those involving triticale formation, are the result of changes of coding and noncoding sequences, regulatory elements, promoter regions, and repetitive sequences. The chromosomal rearrangements arising out of wheat × triticale or wheat × rye crosses have the potential to result in novel gene diversity affecting different agronomic characteristics, as well as processing and nutritional characteristics suggesting that many of the chromosomal changes may prove useful in the breeding improvement of the wheat crop.

## Figures and Tables

**Figure 1 ijms-21-08450-f001:**
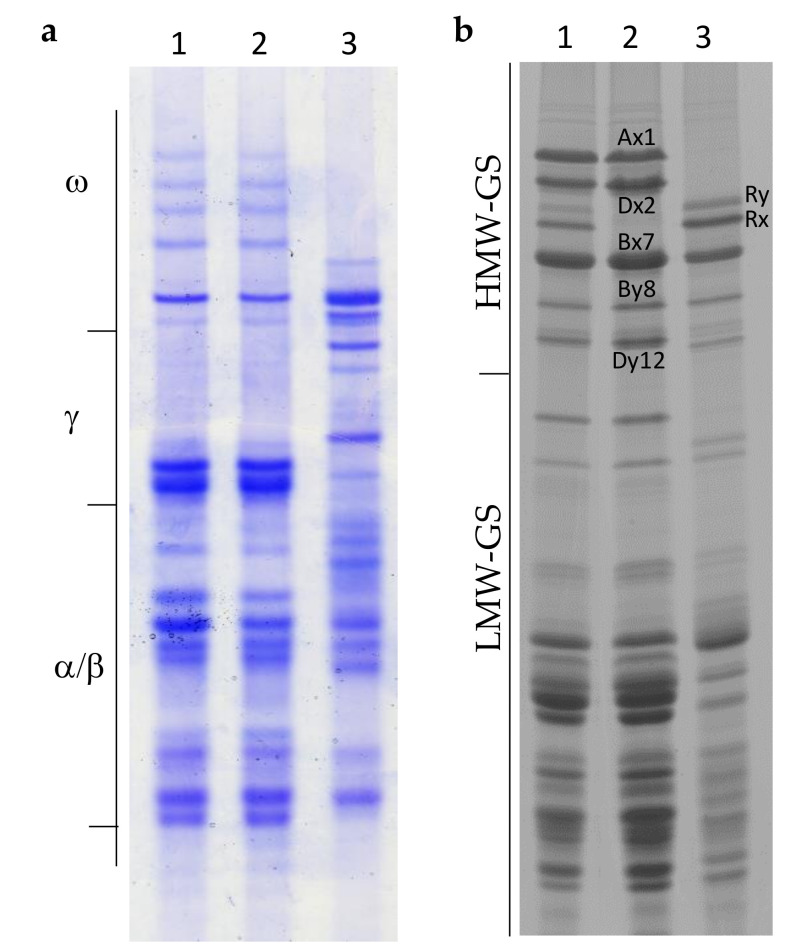
Comparison of gliadin (**a**) and glutenin (**b**) composition of N11, N11^XY7^, and XY7: (**a**) Acid polyacrylamide gel electrophoresis (A-PAGE) separation of gliadins, (**b**) SDS-PAGE separation of glutenins. Lane 1, N11^XY7^; lane 2, N11; lane 3, XY7.

**Figure 2 ijms-21-08450-f002:**
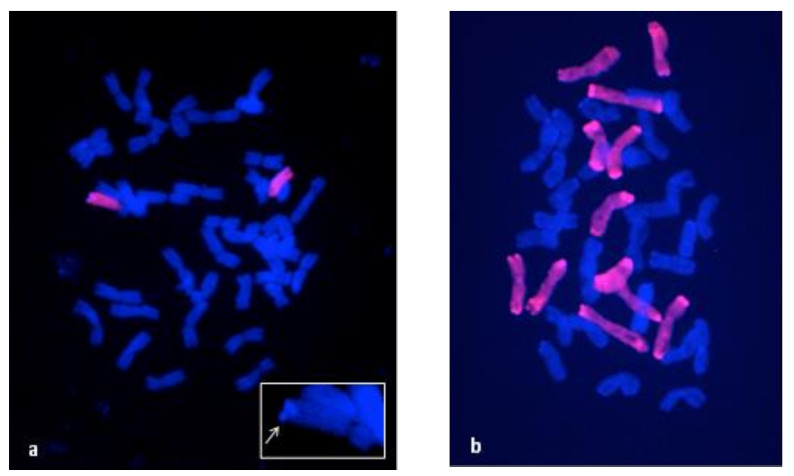
Genomic in situ hybridization (GISH) with total ryeCy3 labeled DNA as probe (red) on N11^XY7^ (**a**) and XY7 (**b**) metaphase spreads. DNA is counterstained with DAPI. Rye chromosomes or chromosome arms are red. In the box a telomeric DAPI band present in the rye chromosome arm is highlighted by a white arrow.

**Figure 3 ijms-21-08450-f003:**
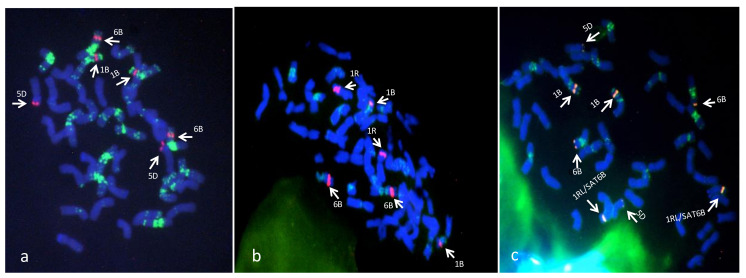
Double target FISH analysis with rDNA pTa71 probe (red) and (GAA)_7_ oligonucleotides (green) on N11 (**a**), XY7 (**b**) and N11^XY7^ (**c**) metaphase spreads. Chromosome rDNA localization is indicated by arrows.

**Figure 4 ijms-21-08450-f004:**
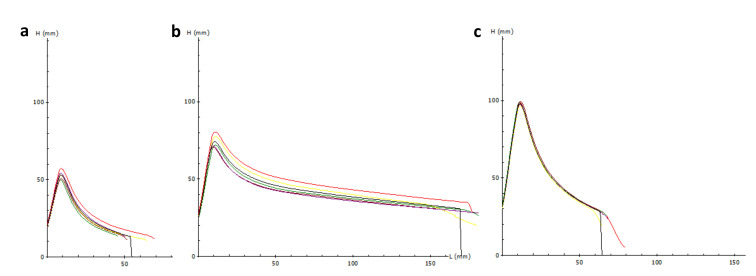
Alveograph profiles of XY7 (**a**), N11^XY7^ (**b**), and (**c**) N11 relative to the 2017–2018 growing season. The colored curves indicate technical replicates.

**Table 1 ijms-21-08450-t001:** Mean of qualitative parameters determined in two different seasons (Y1: 2016–2017 and Y2: 2017–2018).

					Chopin Alveograph				Brabender Farinograph				
	TW (kg/hL)	Hardness	PC % d.m.	GI	P (mm)	L (mm)	P/L	W (J × 10^−4^)	Water Abs (%)	Dough Development Time (min)	Dough Stability (min)	Degree of Softening (BU)	FQN (mm)
**N11**	82.6 ^a^	89	10.6 ^b^	86.5 ^a,b^	100 ^a^	80 ^b^	1.33	231 ^b^	60.9 ^a^	2.8 ^b^	5.3 ^b^	62 ^b^	75.5 ^b^
**XY7**	69.2 ^b^	100	11.0 ^b^	64 ^b^	60 ^b^	55 ^b^	1.09	102 ^c^	54.1 ^b^	1.4 ^b^	3.6 ^b^	119 ^a^	49.5 ^b^
**N11^XY7^**	80.1^a^	96	14.2 ^a^	99.5 ^a^	83 ^a,b^	175 ^a^	0.48	466 ^a^	60.8 ^a^	9.7 ^a^	11.8 ^a^	51 ^b^	171.0 ^a^
**p**	0.04	0.79	0.01	0.016	0.022	0.007	0.13	0.001	0.0029	0.009	0.011	0.015	0.0042
**sem**	0.95	11.69	0.37	3.89	4.77	10.15	0.22	7.02	0.46	0.77	0.81	7.69	8.63
**Y1**	76.5	106	12.1	86.7	83	95	1.09	265	58.4	4.4	6.3	83	92.0
**Y2**	78.0	83	11.7	80.0	79	111	0.84	268	58.7	4.8	7.4	71	103.3
**p**	0.81	0.01	0.80	0.68	0.83	0.78	0.57	0.98	0.93	0.91	0.77	0.70	0.84
**sem**	4.13	3.40	1.14	10.47	11.99	36.89	0.28	106.79	2.27	2.62	2.54	21.42	37.54

Values followed by different letters differ significantly (*p* ≤ 0.05) from one another. TW: test weight; PC: protein content; GI: gluten index; P: dough tenacity; L: dough extensibility; W: dough strength; BU: Brabender Units; FQN: Farinograph quality number; p: probability; sem: standard error of the mean.

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
