# Peer review of "A Cross between Bread Wheat and a 2D(2R) Disomic Substitution Triticale Line Leads to the Formation of a Novel Disomic Addition Line and Provides Information of the Role of Rye Secalins on Breadmaking Characteristics"

_ijms, 2020, doi:10.3390/ijms21228450_

Round 1

Reviewer 1 Report

This manuscript presents an interesting study on the generation of new materials with introgressions that has good quality characteristics. However, its presentation results in some points confusing and difficult to follow for a standard reader. For this reason, some parts in the manuscript should be shortened and clarified for better understanding.

An important point of the work and that should be clarified since it can be extremely confusing, is how and since when was obtained this material. The authors refer to works carried out or published in 2016 (Feng et al. 2016), and later indicate that in 2016-17 the first field trial of the present study was being carried out. One might ask why a material is crossed before knowing what it is. In fact, in Feng et al. 2016, where some of the authors of this paper are co-authors, it is indicated that the cultivar Xiaoyanmai 7 is a bread wheat cultivar, and here they said that it is a substituted triticale. Fact that appears reflected in the work of 2016, but that should have been known previous to perform before doing a field trial with one line selected during 5 generations and derived from a cross plus 4 backcrosses.

On the other hand, in Feng et al. 2016, it is indicated that when species that have the D genome and the R genome are used, the descendants generally keep the R chromosomes and not the D (p. 34). In this work, the XY7 has 6 R-chromosomes plus one D-chromosome, when it is crossed with N11 (a bread wheat) that only has 7 D-chromosomes, it turns out that the descendants lose all the R-chromosomes and only maintain a chimera chromosome formed by 1RL chromosome together with a satellite of chromosome 6B, which also appears duplicated since it is a disomic line. At no time is it said that double haploids have been made, since the generation of the material is scarcely explained. All of this can be extremely confusing for any reader. It is not well understood that the process of obtaining the line is not explained, which in itself would possibly justify an additional paper, and that an excessive amount of cytogenetic questions is included in this manuscript that end up having more weight in the manuscript that the importance of obtaining a new line or cultivar.

On the other hand, the authors compare the values for certain quality parameters of an F5 line derived from the fourth backcross. This point should be clarified, since one part of the text indicates that it is F5 (line 24), but in the Material and Methods section it is said that it is F8 (line 411). This type of comparison should include materials with a common background, in order to establish a clearer difference. The parents serve as controls, but a good comparison should be made between other lines derived from that 4 backcross, with or without the additional chromosome. In this way, it could be determined more clearly whether the changes caused are actually derived from the presence or not of this chromosome and the genes it contains. It must be taken into account that although the genes evaluated correspond to the N11 line, for genomes A and B the chromosomes of N11 and XY7 would be indistinguishable by cytogenetic techniques, and the possible recombination between them would not be taken into account.

Although the measured data are adequate to assess quality, an explanation of the increase in protein content and its stability is lacking. In this regard, the shape and weight of the grain should be measured, since it is essential to rule out incorrect grain-filling, which always increases the protein content. It would also be key to determine if in other lines from this cross that do not present the added chromosome said increase in protein is maintained or not, and if this may or may not be related to some genes present in this additional chromosome.

As the authors themselves recognize, extensibility is mostly related to gliadins and not to HMWG glutenin subunits (lines 349-352); in this case, it is paradoxical that the difference or incorporation of two secalins generate a line with such a notable change in extensibility. To argue that this may be related to the increase in protein, without observing qualitative changes in the proteins related to this property, should be qualified. If this is so, what is the point of the presence or not of secalins? Why this line, which is basically the same as its parent N11, has different viscoelastic properties?

Table 1 should be corrected, since abbreviations have been used that generate confusion. The tenacity is named as P, but the probability is also named as P. In addition, given that a field trial with two years and three repetitions has been carried out, the ANOVA is essential, since the important thing is that the differences observed between the genotypes are not due to the year or the repetition, ruling out one high environmental influence that would render the rest of the study meaningless.

Figure 1 should be done again, it is clearly seen that part b is a photocomposition. It just doesn't make sense to do this on a gel with just three lanes. I would recommend its realization again.

Species and genera are discussed in line 59 of the manuscript, this should be clearer. Dasypyrum villosum is a species, Aegilops is a genus...

The conclusions of the paper should be rewritten, in the current version they are a new discussion. It does not make any sense to include references in a conclusion.

Author Response

Q1: An important point of the work and that should be clarified since it can be extremely confusing, is how and since when was obtained this material. The authors refer to works carried out or published in 2016 (Feng et al. 2016), and later indicate that in 2016-17 the first field trial of the present study was being carried out. One might ask why a material is crossed before knowing what it is. In fact, in Feng et al. 2016, where some of the authors of this paper are co-authors, it is indicated that the cultivar Xiaoyanmai 7 is a bread wheat cultivar, and here they said that it is a substituted triticale. Fact that appears reflected in the work of 2016, but that should have been known previous to perform before doing a field trial with one line selected during 5 generations and derived from a cross plus 4 backcrosses.

R1: When the work started,  simultaneously to the biochemical work we begun the crossing and backcrossing of XY7 with N11 in order to evaluate the effect of the two novel subunits on the rheological characteristics of the doughs. We tought that XY7 was a bread wheat variety as reported in the paper of Wang et al (1993) and we didn't have any particular  problem with the crosses between XY7 and N11. We have added the following  “We must point out that the backcrosses began before we realized that XY7 was not a wheat, as resulted from its chromosomal composition” (lines 105-106

Q2: On the other hand, in Feng et al. 2016, it is indicated that when species that have the D genome and the R genome are used, the descendants generally keep the R chromosomes and not the D (p. 34). In this work, the XY7 has 6 R-chromosomes plus one D-chromosome, when it is crossed with N11 (a bread wheat) that only has 7 D-chromosomes, it turns out that the descendants lose all the R-chromosomes and only maintain a chimera chromosome formed by 1RL chromosome together with a satellite of chromosome 6B, which also appears duplicated since it is a disomic line. At no time is it said that double haploids have been made, since the generation of the material is scarcely explained. All of this can be extremely confusing for any reader. It is not well understood that the process of obtaining the line is not explained, which in itself would possibly justify an additional paper, and that an excessive amount of cytogenetic questions is included in this manuscript that end up having more weight in the manuscript that the importance of obtaining a new line or cultivar.

R2: This is what should have happened in the first crosses made to obtain the variety.XY7, very likely a triticale was crossed with a bread wheat and the D-genome chromosomes were substituted by the R-genome chromosomes. Crossing repeatedly with N11 we have reverted the situation eliminating the R chromosomes and reintroducing the D-genome chromosomes.  We haven’t made double haploids but we backcrossed XY7 four times with N11

Q3: On the other hand, the authors compare the values for certain quality parameters of an F5 line derived from the fourth backcross. This point should be clarified, since one part of the text indicates that it is F5 (line 24), but in the Material and Methods section it is said that it is F8 (line 411). This type of comparison should include materials with a common background, in order to establish a clearer difference. The parents serve as controls, but a good comparison should be made between other lines derived from that 4 backcross, with or without the additional chromosome. In this way, it could be determined more clearly whether the changes caused are actually derived from the presence or not of this chromosome and the genes it contains. It must be taken into account that although the genes evaluated correspond to the N11 line, for genomes A and B the chromosomes of N11 and XY7 would be indistinguishable by cytogenetic techniques, and the possible recombination between them would not be taken into account.

R3: We have made an error which has been corrected. The  line was a BC4F8 and after four generations of backcrossing, N11XY7 will have a genetic composition of 97% similarity with the recurrent parent N11.

Q4: Although the measured data are adequate to assess quality, an explanation of the increase in protein content and its stability is lacking. In this regard, the shape and weight of the grain should be measured, since it is essential to rule out incorrect grain-filling, which always increases the protein content. It would also be key to determine if in other lines from this cross that do not present the added chromosome said increase in protein is maintained or not, and if this may or may not be related to some genes present in this additional chromosome.

R4: The line N11XY7  had high protein content over the two years and genes for high protein content have been reported on chromosome 1R of rye. The seeds of the N11XY7 line had a test weight not significantly different from N11 and weren't shriveled  

Q5: As the authors themselves recognize, extensibility is mostly related to gliadins and not to HMWG glutenin subunits (lines 349-352); in this case, it is paradoxical that the difference or incorporation of two secalins generate a line with such a notable change in extensibility. To argue that this may be related to the increase in protein, without observing qualitative changes in the proteins related to this property, should be qualified. If this is so, what is the point of the presence or not of secalins? Why this line, which is basically the same as its parent N11, has different viscoelastic properties?

R5: We have discussed this point on lines 398 to 409 and suggested that the change we observe in the alveographic parameters could be caused also by the presence of the extra cysteine residue present in XY7, (eight cysteine residues rather than the seven normally present in y-type subunits in bread wheat), with the additional cysteine located within the C-terminal domain upstream of the one normally present in bread wheat. Results have been obtained that show that additional cysteines not always influence positively the bread-making quality of flour, as in the case of the 1Dx5 subunit. On the basis of the rheological data obtained we can hypothesize that the additional cysteine present in the 1Ry subunit could contributes to modulate differently the size distribution of glutenin polymers, producing a dough with a very large extensibility.

Q6: Table 1 should be corrected, since abbreviations have been used that generate confusion. The tenacity is named as P, but the probability is also named as P.

R6: You are right, but Alveograph tenacity is universally expressed by P. We decided to write in Italics the P of probability to avoid confusion

Q7: In addition, given that a field trial with two years and three repetitions has been carried out, the ANOVA is essential, since the important thing is that the differences observed between the genotypes are not due to the year or the repetition, ruling out one high environmental influence that would render the rest of the study meaningless.

R7: the ANOVA was carried out and the results are summarized in table 1. The differences between the two experimental years were not significant for all the parameters analyzed.

Q8: Figure 1 should be done again, it is clearly seen that part b is a photocomposition. It just doesn't make sense to do this on a gel with just three lanes. I would recommend its realization again.

R8: We have redone the Fig.1

Q9: Species and genera are discussed in line 59 of the manuscript, this should be clearer. Dasypyrum villosum is a species, Aegilops is a genus...

R9: We have used the word also for Dasypirum

Q10: The conclusions of the paper should be rewritten, in the current version they are a new discussion. It does not make any sense to include references in a conclusion.

R10: We have reduced the conclusion and removed the references.

Reviewer 2 Report

I hope that all the comments made will allow the authors to more clearly present their unique results in wheat breeding, and make the paper interesting for a wide range of readers.

Author Response

Introduction

Q1: The Introduction should include background and rationale for the work. The last paragraph should contain a summary of the main results and conclusions.

However, the authors in this section included a large fragment (lines 81 to 124) of a discussion of the literature data on the quality of gluten and glutenin proteins. It is advisable to move these discussions to the Discussion section.

At the same time, the main motives that initiated this study and the description of the pedigrees (available data) of the parental forms used with an emphasis on the studied traits (qualitative and quantitative composition of storage proteins and their relationship with baking qualities) are absent or are drowned in the redundant information presented in the lines 81-124. All this does not allow us to correctly assess the practical and scientific value of the work done and the significance of the results obtained.

R1: We have modified the introduction according to the suggestion of the reviewer

Q2: In addition, the same sample of triticale XY7 appears in the manuscript under different nameand  give more information s:

  1. 2D (2R) disomic substitution triticale: in the ‘Title’, ‘Abstract’, and ‘Materials and method’ line 403;
  2. wheat-rye secondary substitution line: line 130;
  3. a bread wheat XY7: line 136;
  4. a a bread wheat XY7: line 141.

All this distracts from the text and makes it difficult to perceive information.

R2: We have uniformed the designation of the XY7 line in: 2D(2R) disomic substitution triticale line

Q3: Discussion.

The objects of this study were both the structure of storage proteins and their relationship with baking qualities in commercial wheat varieties, triticale and a new form obtained on their basis, as well as chromosomal rearrangements in the genomes of parents and a in new commercial form. It is advisable in the section 'Discussion' to use subtitles to discuss each of these two directions of the research.

R3: We have added another section in the discussion dedicated to the role of the protein and differences in HMW secalins that could influence rheological properties of N11XY7

Q4: Conclusion.

Unfortunately, this conclusion is a continuation of the discussion section. However, in this section would be like to see a brief summary of the work done and clear conclusions drawn on the basis of the results obtained, which can be used as the basis for new methods of breeding such cereals as bread wheat and triticale.

R4: We have reduced the conclusions

Round 2

Reviewer 1 Report

In science, each author's background clearly influences any study he or she conducts. In my opinion, based on my previous comments, some things in this study should have been done differently. But of course this is my opinion, which is also influenced by my own background.

In any case, I think the authors have addressed most of the problems and reviewed the MS satisfactorily. I positively value the effort made by the authors for the revision of this manuscript. Now, the objective and the process of obtaining these materials are clearer and more understandable for a standard reader.